# Efficient Analysis of Small Molecules via Laser Desorption/Ionization Time-of-Flight Mass Spectrometry (LDI–TOF MS) Using Gold Nanoshells with Nanogaps

**DOI:** 10.3390/nano14010025

**Published:** 2023-12-21

**Authors:** Noori Kim, Yoon-Hee Kim, Gaon Jo, Jin Yoo, Seung-min Park, Bong-Hyun Jun, Woon-Seok Yeo

**Affiliations:** 1Department of Bioscience and Biotechnology, Konkuk University, Seoul 05029, Republic of Korea; bobbysen@naver.com (N.K.); yoonhees@konkuk.ac.kr (Y.-H.K.); jogaon883@gmail.com (G.J.); jin4261624@gmail.com (J.Y.); 2Department of Urology, Stanford University School of Medicine, Stanford, CA 94305, USA; sp293@stanford.edu; 3School of Chemistry, Chemical Engineering and Biotechnology, Nanyang Technological University, Singapore 639798

**Keywords:** matrix-assisted laser desorption/ionization time-of-flight mass spectrometry, gold nanoshell, nanogap, surface-assisted laser desorption/ionization, silica core nanoparticle, small-molecule analysis

## Abstract

Matrix-assisted laser desorption/ionization time-of-flight mass spectrometry (MALDI–TOF MS) is a commonly used technique for analyzing large biomolecules. However, the utilization of organic matrices limits the small-molecule analysis because of the interferences in the low-mass region and the reproducibility issues. To overcome these limitations, a surface-assisted laser desorption/ionization (SALDI), which utilizes nanostructured metallic surfaces, has been developed. Herein, a novel approach for SALDI–MS was proposed using silica@gold core–shell hybrid materials with a nanogap-rich shell (SiO_2_@Au NGS), which is an emerging material due to its excellent heat-generating capabilities. The gold shell thickness was controlled by adjusting the concentration of gold precursor for the growth of gold nanoparticles. SALDI-MS measurements were performed on a layer formed by drop-casting a mixture of SiO_2_@Au NGS and analytes. At the optimized process, the gold shell thickness was observed to be 17.2 nm, which showed the highest absorbance. Based on the enhanced SALDI capability, SiO_2_@Au NGS was utilized to detect various small molecules, including amino acids, sugars, and flavonoids, and the ionization softness was confirmed with a survival yield upon fragmentation. The limits of detection, reproducibility, and salt tolerance of SiO_2_@Au NGS demonstrate its potential as an effective and reliable SALDI material for small-molecule analyses.

## 1. Introduction

Matrix-assisted laser desorption/ionization time-of-flight (MALDI–TOF) mass spectrometry (MS) is widely used for the analysis of large biomolecules, such as proteins and oligonucleotides, using an organic matrix for soft ionization [1,2,3]. Following the pioneering studies of Tanaka et al., who utilized a metal powder dispersed in glycerol [4,5], and Karas and Hillenkamp, who employed a small organic molecule as a matrix [6], various types of organic matrices, including 2,5-dihydroxybenzoic acid, sinapinic acid, α-cyano-4-hydroxycinnamic acid, and 2,4,6-trihydroxyacetophenone, are commonly used for this purpose depending on the analyte type. MALDI–TOF MS is an effective bioanalytical tool for biomolecules because of its simple and fast operating processes, high tolerance to biological contaminants, and the ability to facilitate multiple analyses simultaneously. However, the necessity of using an organic matrix considerably limits its practical application (particularly for small-molecule analyses) because of the interferences in the low-mass region. In addition, the solvents utilized for dissolving the analyte and matrix, the co-crystallization of the matrix and analyte, and even the deposition method strongly affect the shot-to-shot and sample-to-sample reproducibility of this technique. To overcome these limitations, significant efforts have been devoted to developing an organic matrix-free analysis method known as surface-assisted laser desorption/ionization (SALDI), which is primarily based on the use of nanostructured surfaces and inorganic nanoparticles (NPs) instead of the organic matrix [7,8]. Since the pioneering study of Tanaka, who employed 30 nm cobalt NPs dispersed in glycerol for lysozyme analysis [5], various nanomaterials based on metals, including Au, Ag, TiO_2_, Zn, and Pt, have been applied for SALDI–MS [9]. Furthermore, carbon-based materials, such as graphite, carbon nanotubes, graphene oxide (GO), and graphene derivatives, have been actively used for this purpose [10,11,12]. In particular, inorganic NPs, including gold nanoparticles (AuNPs) [13] and silver NPs [14], are widely employed in SALDI–MS because of their advantageous physiochemical properties, photothermal activities, and simple fabrication procedures affording homogeneously controllable morphologies; however, the use of these NPs raises issues such as maintaining colloidal stability during analysis and heat dissipating after laser irradiation owing to their high thermal conductivity. Therefore, several attempts have been made to obtain efficient SALDI substrates by confining the laser-driven energy to nanoscale gap structures. As a result, substrates with closely packed nanostructures fabricated via vapor deposition [15] and droplet deposition [16] were produced. The SALDI efficiency and signal strength of these substrates were strongly correlated with their sub-micrometer and nanometer-scale morphologies, which could be controlled by adjusting the sputtered metal amount and the number of deposition cycles. However, fabricating homogeneous nanogap structures via these methods is difficult owing to the bulk agglomerate formation and uncontrolled interparticle aggregation during the deposition process.

Recently, “gold nanoshells”, core–shell NPs consisting of silica NP cores and AuNP surfaces, were utilized for the first time as a SALDI substrate by Jinrui et al. [17]. Such materials can effectively confine heat by the silica core NPs and abundant “hot spots” distributed across the interparticle junctions of the closely packed AuNPs on the silica surface. The existence of correlations between the SALDI signals of various analytes and the thickness of gold nanoshells were discussed in detail by Mingyi et al. [18]. They prepared three types of gold nanoshells and compared their SALDI efficiencies, which depended on the surface coverage and shell roughness at the nanometer scale. The optimized gold nanoshells with the highest SALDI efficiency have been successfully applied in the mass spectrometry imaging of the mouse brain. In these applications, gold nanogap structures generated “hot spots”, which consisted of the highly localized regions of an intense electromagnetic field due to plasmonic coupling between adjacent nanostructures. Recently, we have prepared gold nanogap shells using silica NPs, which significantly amplified a local electromagnetic field [19,20]. Finely controlled nanogap structures were obtained by increasing the shell thickness, which in turn decreased the average nanogap size from approximately 4 to 1 nm. Although these NPs were utilized as substrates and catalysts for surface-enhanced Raman scattering spectroscopy, they have not been employed for SALDI yet. Herein, we propose an organic matrix-free LDI–MS method using silica@gold core–shell hybrid nanocomposites with nanogap-rich shell (SiO_2_@Au NGS) structure. SiO_2_@Au NGS with various shell thicknesses and well-ordered surface structures were synthesized by modulating the Au precursor concentration during growth.

## 2. Experimental Section

### 2.1. Chemicals

Quercetin hydrate, cellobiose, triethylene glycol, tetraethylene glycol, tetraethyl orthosilicate (TEOS), gold (III) chloride, polyvinylpyrrolidone (PVP, average molecular weight: ~10,000), (3-aminopropyl) trimethoxysilane (APTS), and ascorbic acid were purchased from Sigma–Aldrich (St. Louis, MO, USA). Glutamic acid, mannitol, serine, galactose, histidine, and aqueous ammonium hydroxide (NH_4_OH, 25–28%) were procured from Daejung Chemical & Metals Co., Ltd. (Gyeonggi-do, Republic of Korea). Aspartic acid and acetonitrile were acquired from Junsei Chemical Co., Ltd. (Tokyo, Japan). Kaempferol hydrate and pentaethylene glycol were purchased from the Tokyo Chemical Industry (Tokyo, Japan). Tryptophan, arginine, and glutamine were procured from Samchun Chemicals Co., Ltd. (Gyeonggi-do, Republic of Korea). Absolute ethanol was obtained from Merck (Darmstadt, Germany). GO was purchased from Graphene Laboratories Inc. (Ronkonkoma, NY, USA). Benzyl pyridinium salt was prepared using a method previously reported by Tang et al. [21]. Deionized water was obtained using an AquaMAX Ultra 370 water purification system (Younglin Instruments, Anyang, Republic of Korea).

### 2.2. Synthesis of SiO_2_@Au NGS

AuNPs were prepared using tetrakis (hydroxymethyl) phosphonium chloride (THPC) as a reducing agent. First, a NaOH solution (0.2 M, 1.5 mL) was diluted with deionized water (47.5 mL), followed by the sequential addition of THPC (80%, 12 μL) and gold (III) chloride (50 mM, 1 mL) solutions. The obtained mixture was vigorously stirred for 1 h. AuNP-seed-immobilized silica NPs (SiO_2_@Au) were synthesized using a silica NP template. Silica NPs were prepared via the Stöber process [22]. Briefly, TEOS (1.6 mL) was mixed with ethanol (40 mL), followed by the addition of the NH_4_OH solution (3–5.5 mL) to obtain silica NPs of various sizes. The reaction mixture was stirred vigorously for 1 h at 60 °C and then for 19 h at 25 °C. The produced solution was washed several times with ethanol via centrifugation at 9000× *g*. The resulting silica NPs (15 mg) were modified with amino groups by stirring with the APTS (15.5 μL) and NH_4_OH (10 μL) solutions in a vortex mixer for 12 h. The mixture was washed several times with ethanol via centrifugation at 9000× *g*, and the obtained aminated silica NPs (2 mg) were mixed with the AuNP solution (10 mL) overnight. After washing with deionized water, the resulting dark-brown pellets were dispersed in ethanol. SiO_2_@Au NGS were synthesized through the seed-mediated growth of SiO_2_@Au at different concentrations of the gold precursor solution. Briefly, SiO_2_@Au (10 mg) was dispersed in an aqueous PVP solution and stirred at 500 rpm. During stirring, 200 μL of a 50 mM Au^3+^ aqueous solution and 100 mM ascorbic acid aqueous solution were added simultaneously at 5 min intervals. The addition of the precursor solutions was repeated to achieve a final Au^3+^ concentration of 0.5, 1.0, 1.5, or 2 mM. The obtained mixture was washed with deionized water and redispersed in ethanol.

### 2.3. Characterization

Transmission electron microscopy (TEM) images were captured using JEM-1010 (JEOL, Tokyo, Japan) and JEM-2010 (JEOL, Tokyo, Japan) instruments with acceleration voltages of 80 and 120 kV, respectively. The UV–visible absorption spectra were recorded using a single-beam UV/visible spectrophotometer (U-5100, HITACHI, Tokyo, Japan).

### 2.4. MALDI-TOF MS Analysis of Small Molecules

To prepare the samples for LDI–TOF MS analyses, 100 μM stock solutions of small molecules were produced. The amino acids and sugars were dissolved in distilled water, and oligoethylene glycols and flavonoids were dissolved in ethanol. SiO_2_@Au NGS_0.5–2.0_ (10 μL, 1 mg/mL in ethanol) were mixed with the analyte solutions (5 μL) at various concentrations, and the obtained mixtures (1.5 μL) were pipetted onto stainless-steel 384-well target plates (Bruker Daltonics, Bremen, Germany). The prepared samples were dried under a vacuum at 25 °C and analyzed directly via MS. The MALDI–TOF MS analyses were performed using an Autoflex III MALDI–TOF mass spectrometer (Bruker Daltonics, Germany) equipped with a smart beam laser (λ = 355 nm) serving as an ionization source. All spectra were acquired in a positive mode at an accelerating voltage of 19 kV and a repetition rate of 50 Hz with an average number of shots equal to ~500.

## 3. Results and Discussions

### 3.1. Preparation and Characterization of SiO_2_@Au NGS

The fabrication process of SiO_2_@Au NGS is illustrated in Figure 1. SiO_2_@Au NGS was synthesized in two steps, including the attachment of AuNP seeds to silica NPs and the seed-mediated growth of AuNP seeds. To attach AuNPs to the substrate surface, silica NPs (~178 nm) were aminated with APTS. Then, the aminated silica NPs were mixed with AuNPs (~3 nm) to prepare the AuNP seed-immobilized silica NPs. Subsequently, gold precursor solutions with different concentrations were added to grow AuNP seeds, resulting in SiO_2_@Au NGS. The SiO_2_@Au NGS morphology was controlled by varying the concentration of the Au^3+^ precursor in the reaction mixture during the growth process, as shown in the corresponding transmission electron microscopy (TEM) images (Figure 1a–f). The thickness of the gold shell was averaged from the equivalent spherical diameter of 30 AuNPs that constitute the outer gold shell. By varying the concentration of Au^3+^ precursors as 0, 0.5, 1.0, 1.5, and 2.0 mM, shell thicknesses of each particle were measured as 2.3 ± 0.4, 8.3 ± 1.6, 11.9 ± 2.6, 13.1 ± 2.4, and 17.2 ± 2.0 nm (SiO_2_@Au NGS_0.5–2.0_). Notably, the shell thickness affected not only the nanogap size but also plasmonic absorption properties. Characteristic plasmonic frequencies of the SiO_2_@Au NGS with different Au^3+^ concentrations were observed in their absorption spectra recorded in the ultraviolet (UV)/visible range (Appendix A). When using a higher concentration of the Au^3+^ precursor, we found uncontrolled growth of individual AuNPs, and therefore, we could not obtain NGS materials. In addition, we presume that nanogap structures on the surface would not be afforded at higher concentrations of the Au^3+^ precursor because of the coalescence among AuNPs. To evaluate the laser irradiation absorption properties of various SiO_2_@Au NGS samples during LDI, their absorbances at 355 nm are compared in Figure 1f. The absorbance gradually increases with increasing Au^3+^ concentration from 0 to 1.5 mM, and a slight increase in absorbance is observed at 2.0 mM. These structural and optical characteristics can be used to predict the desorption efficiency during LDI spectrometry using the laser-driven energy for the desorption and ionization of analytes.

### 3.2. Matrix Properties of SiO_2_@Au NGS_0.5~2.0_ Evaluated Using Small Molecules

To utilize the prepared SiO_2_@Au NGS as a matrix for LDI–TOF spectrometry, a layer of SiO_2_@Au NGS was formed by drop-casting a mixture of colloidal SiO_2_@Au NGS and target analytes, and desorption and ionization of these analytes were performed by irradiating the resulting layer with a pulsed laser beam (Figure 1). We examined the feasibility of utilizing SiO_2_@Au NGS_0.5–2.0_ as matrices for LDI–TOF MS analysis and their efficiency by studying various small molecules, including amino acid serine, sugar mannitol, quercetin, and pentaethylene glycol.

The analytes were dissolved in distilled water or ethanol, considering their solubility differences, and the resulting solutions were mixed with the synthesized SiO_2_@Au NGS_0.5_, SiO_2_@Au NGS_1.0_, SiO_2_@Au NGS_1.5_, and SiO_2_@Au NGS_2.0_. AuNP seed-immobilized silica NPs (SiO_2_@Au NGS_0_) produced without seed-mediated gold growth were used as a control. Five representative mass spectra recorded for each of the four analytes are shown in Figure 2, and their detailed peak assignments with peak intensities, signal-to-noise (S/N) ratios, and resolutions are summarized in Appendix A. In general, the adduct ion peaks corresponding to the studied analytes are clearly observed in Figure 2, and their intensities increase with increasing concentration of the gold precursor solution during the SiO_2_@Au NGS synthesis. Except while using quercetin, the peak intensities of SiO_2_@Au NGS_2.0_ are higher than those of SiO_2_@Au NGS_1.5_, although SiO_2_@AuNGS_2.0_ and SiO_2_@AuNGS_1.5_ have a similar absorption peak at 355 nm—by a factor of 4 for serine, 25 for mannitol, and 10 for pentaethylene glycol. We presumed that the surface area of the embedded gold shell nanostructures with nanogaps on the silica surface increased with increasing gold precursor concentration to facilitate the accommodation of the energy supplied by laser irradiation and its subsequent transfer to the analytes, which increased the SALDI efficiency. In addition, the enhanced plasmonic properties of SiO_2_@Au NGS and rapid heating due to the low thermal conductivity of the silica NP core significantly enhanced the SALDI effect. In summary, SiO_2_@Au NGS_2.0_ exhibited the optimal matrix properties among the four gold shell materials, and we further validated the versatility of this material for LDI–MS by analyzing various small molecules and comparing it with other well-known SALDI materials.

### 3.3. Analyses of Small Molecules and Their Mixtures

Next, we analyzed various small molecules using SiO_2_@Au NGS_2.0_ to evaluate the matrix properties of the synthesized material. Four classes of small molecules were examined: amino acids (aspartic acid, glutamic acid, and glutamine), oligoethylene glycols (triethylene glycol and tetraethylene glycol), sugars (mannitol and galactose), and flavonoids (quercetin and kaempferol). Analytical solutions for LDI–MS analysis were prepared using distilled water for amino acids and sugars and ethanol for oligoethylene glycols and flavonoids. As shown in Figure 3, the SALDI spectra of SiO_2_@Au NGS_2.0_ exhibit distinct strong peaks corresponding to the adduct ions of amino acids (a–c), oligoethylene glycols (d,e), sugars (f,g), and flavonoids (h,i) with high intensities and S/N ratios without any significant background interferences (detailed peak assignment is provided in Appendix A). Note that a gold peak is observed at *m*/*z* = 197. Subsequently, mixtures of small molecules were tested to confirm the separation of each analyte from the mixed samples. We prepared mixed samples containing three amino acids (glutamine, histidine, and arginine), three oligoethylene glycols (triethylene glycol, tetraethylene glycol, and pentaethylene glycol), three sugars (galactose, mannitol, and cellobiose), and three molecules from different classes (tryptophan, quercetin, and cellobiose). The mass spectra of each analyte mixture also clearly show major peaks corresponding to the adduct ions of the studied analytes with weak background peaks (Figure 4; for detailed peak assignment, see Appendix A). These results strongly suggest that the nanoengineered gold shells with nanogaps on the silica surface SiO_2_@Au NGS_2.0_ with well-ordered surface structures are a highly effective SALDI material for the analysis of various small molecules and their mixtures.

### 3.4. Limits of Detection (LODs) and Reproducibility

LODs were measured for three analytes (glutamine, cellobiose, and kaempferol) to evaluate the practical feasibility of the SiO_2_@Au NGS_2.0_-based LDI technique. The LODs obtained at S/N ratios higher than six were 37.5 pmole for glutamine, 7.5 fmole for cellobiose, and 75 fmole for kaempferol (Figure 5). These values are comparable to those determined in previous studies using gold-based materials [13,23]. We also investigated the reproducibility of the fabricated material via multiple mass determinations (n = 25) for cellobiose (100 pmole in 1 μL on the target plate). The calculated relative standard deviation of 18.6% was comparable to or better than the values obtained in other studies using AuNPs (Appendix A) [24].

### 3.5. Comparison of AuNP, GO, and SiO_2_@Au NGS Matrices

The superiority of SiO_2_@Au NGS_2.0_ as a SALDI matrix over other commonly used inorganic matrices, AuNPs and GO was verified by comparing their SALDI properties using four analytes: kaempferol, cellobiose, triethylene glycol, and serine. The mass spectra obtained for AuNPs, GO, and SiO_2_@Au NGS_2.0_ are shown in Figure 6, and a detailed peak assignment is provided in Appendix A. The GO spectrum exhibited kaempferol peaks with the highest intensity compared to the intensities of the three compounds but with very poor resolution, whereas the other three analytes were negligibly observed. AuNPs showed peaks with decent intensities and S/N ratios for four analytes. Particularly for kaempferol analysis, peak intensities, S/N ratios, and resolutions obtained using AuNPs were comparable to or better than those obtained using SiO_2_@Au NGS_2.0_. For cellobiose, triethylene glycol and serine, SiO_2_@Au NGS_2.0_ generated the most intense analyte peaks with higher S/N ratios and resolution with fewer background signals than AuNPs. We also examined the background signals in a higher range up to ~500 for kaempferol and cellobiose, and we found AuNPs afforded more background signals than SiO_2_@Au NGS_2.0_ (Appendix A). Furthermore, we examined the applicability of SiO_2_@Au NGS_2.0_ as a SALDI matrix using benzylpyridinium (BP) salt, which has been adopted as a “chemical thermometer” to investigate the ion desorption efficiency and internal energy transfer by measuring a survival yield (SY) [21]. BP is fragmented via laser irradiation during the LDI process to produce benzylic carbocations (Figure 7a); therefore, the SY of BP is an important parameter characterizing the SALDI efficiency and ionization “softness”. The SY was calculated by dividing the ion intensity of the intact BP precursor by the sum of the intact BP and fragmented product intensities after LDI–MS measurements. Five SALDI analyses were conducted for BP using AuNPs, GO, and SiO_2_@Au NGS_2.0_ under the same experimental conditions. The obtained representative spectra are shown in Figure 6b. The average SY values determined from the five mass spectra (0.63 for AuNPs, 0.41 for GO, and 0.64 for SiO_2_@Au NGS_2.0_) imply that the ion desorption and internal energy transfer efficiency of AuNP is comparable to SiO_2_@Au NGS_2.0_; however, the standard deviation for SiO_2_@Au NGS_2.0_ (0.04) much better than those for other materials (0.16 for AuNPs and 0.20 for GO).

Another major issue encountered during the MALDI analysis is the signal suppression for analytes with salt contaminants (typically 150 mM NaCl) [25]. Therefore, SALDI materials must maintain colloidal stability during analysis (particularly when using salt-containing samples) and be tolerant to high salt concentrations [26]. As shown in Appendix A, SiO_2_@Au NGS_2.0_ showed excellent colloidal stability in NaCl solutions with concentrations up to 1 M, whereas AuNPs lost their stability at 50 mM or higher NaCl concentration. We also tested two analytes, histidine and mannitol, to investigate the salt tolerance of SiO_2_@Au NGS_2.0_. The analytes were prepared in NaCl solutions at concentrations ranging from 0.1 to 1 M and then examined using SiO_2_@Au NGS_2.0_. The SALDI spectra of histidine clearly show the main peaks of the sodium adduct ([M+Na]^+^) at *m*/*z* = 178.06 and sodiated sodium adduct ([M+2Na–H]^+^) at *m*/*z* = 200.56 (Figure 8a). For mannitol, a sodium adduct peak (*m*/*z* = 205.07, [M+Na]^+^) was observed as the main peak (Figure 8b). For both analytes, SiO_2_@Au NGS_2.0_ generated sodium adduct peaks as the main peaks at NaCl concentrations up to 1 M, indicating a high salt tolerance of the SALDI material; however, the peak intensities decreased with increasing salt concentration. For comparison, we also performed the SADLI analysis with AuNPs in high salt concentration. Unlike SiO_2_@Au NGS_2.0_, most SALDI spectra showed highly intense gold background peaks (*m*/*z* = 197), whereas analyte peaks were observed with low intensity and hardly observable at higher salt concentrations, which can account for the low colloidal stability of AuNPs compared to SiO_2_@Au NGS_2.0_ (Appendix A).

Taken all together, the results and discussion above in terms of peak intensities, resolutions, S/N ratios, background signals, SY values, colloidal stability, and salt tolerance clearly indicate the superiority of SiO_2_@Au NGS_2.0_ as a SALDI matrix over other commonly used inorganic matrices, AuNPs and GO.

## 4. Conclusions

In summary, nanoengineered gold shells with nanogaps on the silica surface were utilized for the SALDI–MS analysis of small molecules. As the gold shell thickness increased, the plasmonic absorption of the SiO_2_@Au NGS increased gradually, while maintaining its well-ordered nanogap structure. In addition, the signal intensity and S/N ratio of the SALDI-MS peaks also exhibited a similar correlation with the gold shell thickness of SiO_2_@Au NGS. The phenomenon is presumed to be because of the high UV absorbance, which is advantageous for heat generation and contributes to the desorption of the analyte. Based on the excellent SALDI performance of SiO_2_@Au NGS_2.0_, various small molecules, including amino acids, sugars, flavonoids, and their mixtures, were investigated. This material demonstrated high reproducibility and salt tolerance, and its LODs were comparable to or better than those of other SALDI materials. Therefore, the material fabricated in this study can potentially replace the matrix-free LDI–MS analytical tools currently used by researchers.

## Data Availability

The data presented in this study are available on request from the corresponding author.

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
