# Peer review of "Efficient Analysis of Small Molecules via Laser Desorption/Ionization Time-of-Flight Mass Spectrometry (LDI–TOF MS) Using Gold Nanoshells with Nanogaps"

_nanomaterials, 2023, doi:10.3390/nano14010025_

Round 1
Reviewer 1 Report (New Reviewer)
Comments and Suggestions for Authors The manuscript shows a trial of gold nanoparticle-embedded silica nanoparticles (SiO2@Au NGS) for applications of a kind of mass spectrometry (LDI–TOF MS). The concept is clear and, however, the result does NOT indicate the significance of the SiO2@Au NGS especially compared to gold nanoparticles (AuNPs). In addition, there are a few unclear points in the results. Thus, I recommend publishing this manuscript in this journal after following revisions.
- The performance of AuNP seems comparable to that of [email protected]. Especially, the S/N ratios of AuNP in the case of Kaempferol analyte are superior to that of [email protected], although the case of [email protected] show the best performance in the other three S/N ratios. The authors should describe the difference performance in the four analytes between AuNP and [email protected], as well as the similar SY value between the two. From the results and discussion in the manuscript, it is difficult to judge that [email protected] is an excellent SALDI matrix more than AuNP.
-Salt tolerance can show the significance of [email protected] matrix against the others. The result of AuNP matrix in high salt concentration should be made and compared to that in Figure 8.
- There is no “linear” relationship between signal obtained and shell thickness. SiO2@AuNGS1,5 seems fine particle and has a similar absorption peak at 355 nm to [email protected]. The author should discuss the large difference in performance as SALDI matrix between [email protected] and [email protected]. How about salt tolerance in the case of [email protected]? As Fig. S3, full spectra of the four SiO2@AuNGS should be shown from UV region to visible region in Fig. S1.
Author Response
Please see the attachment.

Reviewer 2 Report (New Reviewer)
Comments and Suggestions for Authors
MALDI–TOF MS is normally used to analyze larger biomolecules using an organic matrix for ionization. Matrices are for instance DHBA, sinapinic acid and α-CHCA.
Nevertheless, matrices have the shortcomings that interferences with analytes in the low mass region is present and that the solvents and the co-crystallization of the matrix and analyte can affect the reproducibility of this technique. +
In this study, the author tried to solve this problem by a matrix-free technique called surface-assisted laser desorption/ionization (SALDI). For this, they used silica@gold core-shell hybrid materials with nanogaps-rich shell (SiO2@Au NGS).
The authors could show in several experiments that small molecules like amino acids or sugar become visible free of interferences. They varied gold shell thickness and salt concentration. They stated that this optimized material can potentially replace the matrix-free LDI–MS analytical tools currently used for analysis of small molecules.
The experiments were carried out by the authors in a sensible and comprehensible manner. The results were well discussed and contrasted with similar techniques.
I have no further comments.
Author Response
We are truly grateful for the comment of the reviewer and believe that the revised manuscript will make an exciting contribution to SALDI analysis of various small molecules. Thank you again for your interest in our work.
This manuscript is a resubmission of an earlier submission. The following is a list of the peer review reports and author responses from that submission.
Round 1
Reviewer 1 Report
Comments and Suggestions for Authors
Review on
Efficient analysis of small molecules via laser desorption/ionization time-of-flight mass spectrometry (LDI–TOF MS) using gold nanoshells with nanogaps
In this article, an interesting SALDI-MS method has been developed. The authors created a novel type of nanoparticle to improve the applicability of the SALDI-MS method. The research is well designed they used many different types of compounds (flavonoids, amino acids, oligoethylene glycols, and carbohydrates) to test and validate the method.
The results are promising, but the evaluation of the results is not complete. Therefore, I suggest a major revision before acceptance.
My questions and suggestions are the following:
1: The intensity of the compounds has been used to evaluate the method, in some cases the signal-to-noise ratio and the resolution have been mentioned. These parameters are more important than the intensity when comparing the developed nanoparticles with other methods. The intensity is acceptable in the article as the same instrument was used, but the signal-to-noise ratio and resolution of the peaks should at least be added to the supporting information for all peaks in all spectra (as given in the supporting information (m/z and intensity).
2 In the part "Comparison of AuNP, GO and SiO2@Au NGS matrices" the spectra contain more peaks than expected and many of them are not assigned, it would be worth comparing the background spectra in a higher range of e.g. 100-1000. It seems that the use of AuNPs results in more background peaks.
3 In the part "Comparison of AuNP, GO and SiO2@Au NGS matrices" I really missed the comparison of the resolution of the peaks. It is one of the most important parameters of the peaks, it is mentioned in the text, but further evaluation is needed to prove the applicability of the developed nanoparticles. The use of GO seems to be problematic due to the low resolution.
4. The SY values for the AuNP and SiO2@Au NGS matrices are similar, but comparisons are only possible if the deviation or (better) confidence interval is given.
5 In Figure 8 the peak at m/z 205.01 was assigned as mannitol. The simulated m/z is 205.068259. The difference is large, the exact elemental composition cannot be determined or even estimated. The accuracy of the measurements and the specifications should be compared. The low accuracy is due to the instrument, this is the result of the matrix?
The figures should be placed in the order in which they first appear. For example, Figure 7 is on page 9 and in the text it is on page 10 after Figure 8.
Other comments.
Page 1, line 22 (Abstract) "The ion desorption efficiency was confirmed with a survival yield upon fragmentation."
The sentence is incorrect. Fragmentation can be studied by the survival yield method, but desorption cannot.
Scheme 1 is useful for readers to understand the procedure, but at the very beginning of the article this positive feature does not prevail. This diagram is more of a summary than an introduction.
On page 6, line 196, there are no molecular ions in the spectra, they are adduct ions.
In general, sodium adduct is written instead of sodiated adducts or sodium ion adducts. Also, the two sodium adduct is incorrect as it suggests doubly charged ions. It is a sodiated sodium salt.
Reviewer 2 Report
Comments and Suggestions for Authors
General Comments
The manuscript presents a comprehensive study on LDI-TOF MS using gold nanoshells with nanogaps, specifically focusing on SALDI-MS analysis. Overall, the paper is well-structured, and the methodology is sound. The results are presented clearly, and the discussion is insightful. However, there are some minor issues that need to be addressed to improve the clarity and completeness of the paper.
Title and Abstract
- Clarification on NGS: The term "NGS" appears in the title and abstract but is not clearly defined. It would be beneficial for the reader to understand what NGS stands for without having to delve into the main content.
Methodology
- Section 2.4 - Equipment Details: The manuscript mentions the use of MALDI-TOF (Bruker Daltronics) equipment but does not specify the laser wavelength. Could you please clarify if the laser wavelength is 355 nm?
Results and Discussion
-
Section 3.1 - Thickness Measurement: The paper discusses various thicknesses like 2.3, 8.3, 11.9, etc., but it is unclear how these measurements were determined. Could you please elaborate on the methodology used for these measurements and the error tolerance?
-
Effect of Larger NGS Values: The paper currently focuses on NGS0 to NGS2.0. It would be interesting to discuss the potential effects or implications of using larger values like NGS5.0.
Summary
The paper is well-written and contributes valuable insights into the field of nanomaterials and SALDI-MS analysis. However, addressing the above-mentioned minor issues will enhance the paper's clarity and completeness.
Reviewer 3 Report
Comments and Suggestions for Authors
This paper describes LDI–TOF MS analysis using silica nanoparticles modified with gold nanoparticles. Although favorable results are obtained, advantage of this method is not clear, because the results are generally comparable to Au nanoparticles method. Phenomenon characteristic to the use of this new nanoparticles is not described. This work is premature, and this referee cannot recommend publication in Nanomaterials.
Round 2
Reviewer 1 Report
Comments and Suggestions for Authors
the authors responded to all my questions and modified the manuscript accordingly.
Author Response
We are truly grateful for the constructive comments of the referee and believe that this work will make an exciting contribution to the organic matrix-free DLI-TOF MS analysis.
Reviewer 3 Report
Comments and Suggestions for Authors
Unfortunately, this referee still cannot understand experimental advantages and characteristic aspects of this method over Au NP method, despite the following notes in reply letter and in text.
As shown in paragraph 3.4, 'Comparison of Au NP, GO, and SiO2@Au NGS Matrices' (Pages 8-9), MS spectra for four types of small molecule analytes (Figure 5) and the fragmentation patterns for BP (Figure 6) showed that SiO2@Au NGS exhibited excellent SALDI matrix properties during small-molecule analysis.
Line 277. “the mass spectra obtained for SiO2@Au NGS2.0 are more distinct than those recorded for AuNPs and GO”.
Line 282. “SiO2@Au NGS2.0 exhibited higher peak intensities and resolution with less background signals than AuNPs”.
Author Response
In general, SALDI formats are categorized into nanostructured surfaces and nanoparticles. Although nanostructured surfaces are beneficial, since this format presents better robustness and can be manufactured as a target plate, the use of nanoparticles is advantageous over nanostructured surface format in terms of their facility for optimizing analysis protocols. Particularly, gold nanoparticles (AuNPs) are most commonly used as they have advantages over other materials due to their unique optical properties of high molar absorptivity. However, the use of AuNPs raises issues such as maintaining colloidal stability during analysis (particularly, when using salt-containing samples) and heat dissipating after laser irradiation owing to their high thermal conductivity. To overcome these issues, AuNP-concentrated surfaces of patterned nanostructures can be introduced as an ionization platform. The gold nanoshells as in our study can effectively confine heat by the silica core NPs (silica has low thermal conductivity) and abundant hot spots distributed across the interparticle junctions (nanogaps) of the closely packed AuNPs on the silica surface. In addition, we can finely control nanogap size from approximately 4 to 1 nm and utilized resulting materials for SALDI analysis. We strongly believe that the presented data in the manuscript clearly indicate the potential of our material as an effective and reliable SALDI material for small-molecule analyses.
Round 3
Reviewer 3 Report
Comments and Suggestions for Authors
This referee understood that the present NP can exhibit interesting properties. However, experimental data showing the property is necessary because this is a scientific paper.